# Endogenous Signaling Molecule Activating (ESMA) CARs: A Novel CAR Design Showing a Favorable Risk to Potency Ratio for the Treatment of Triple Negative Breast Cancer

**DOI:** 10.3390/ijms25010615

**Published:** 2024-01-03

**Authors:** Mira Ebbinghaus, Katharina Wittich, Benjamin Bancher, Valeriia Lebedeva, Anijutta Appelshoffer, Julia Femel, Martin S. Helm, Jutta Kollet, Olaf Hardt, Rita Pfeifer

**Affiliations:** 1Miltenyi Biotec B.V. & Co. KG, 51429 Bergisch Gladbach, Germany; mira.ebbinghaus@miltenyibiotec.de (M.E.); katharina.wittich@miltenyibiotec.de (K.W.); benjamin.bancher@miltenyibiotec.de (B.B.); valeriia.lebedeva@miltenyibiotec.de (V.L.); anijutta.appelshoffer@miltenyibiotec.de (A.A.); julia.femel@miltenyibiotec.de (J.F.); martin.helm@miltenyibiotec.de (M.S.H.); jutta.kollet@miltenyibiotec.de (J.K.); 2School of Applied Biosciences and Chemistry, HAN University of Applied Sciences, 6525 EM Nijmegen, The Netherlands

**Keywords:** CAR T cells, solid tumors, triple-negative breast cancer, alternative CAR signaling, endogenous CAR signaling mechanisms, T cell persistence, T cell exhaustion, in vivo studies, immunofluorescent imaging

## Abstract

As chimeric antigen receptor (CAR) T cell therapy continues to gain attention as a valuable treatment option against different cancers, strategies to improve its potency and decrease the side effects associated with this therapy have become increasingly relevant. Herein, we report an alternative CAR design that incorporates transmembrane domains with the ability to recruit endogenous signaling molecules, eliminating the need for stimulatory signals within the CAR structure. These endogenous signaling molecule activating (ESMA) CARs triggered robust cytotoxic activity and proliferation of the T cells when directed against the triple-negative breast cancer (TNBC) cell line MDA-MB-231 while exhibiting reduced cytokine secretion and exhaustion marker expression compared to their cognate standard second generation CARs. In a *NOD SCID Gamma (NSG)* MDA-MB-231 xenograft mouse model, the lead candidate maintained longitudinal therapeutic efficacy and an enhanced T cell memory phenotype. Profound tumor infiltration by activated T cells repressed tumor growth, further manifesting the proliferative capacity of the ESMA CAR T cell therapy. Consequently, ESMA CAR T cells entail promising features for improved clinical outcome as a solid tumor treatment option.

## 1. Introduction

Despite continuous therapy advancements, there remains a lack of suitable treatment options against a number of solid tumor malignancies [1]. One of them is triple-negative breast cancer (TNBC), which represents 15–20% of all breast cancer cases and is considered to be the most aggressive breast cancer form with the poorest prognosis [2,3]. In contrast to other subtypes, TNBC is insensitive to the commonly used breast cancer biomarker-targeting therapies, as it is hormone receptor-negative and lacks human epidermal growth factor receptor 2 (HER2) overexpression. Therefore, treatment options are predominately limited to surgical interventions, chemotherapy, and radiotherapy. Notably, despite commonly being sensitive to chemotherapy, TNBC is highly prone to relapse within the first three years after diagnosis [2,4], illustrating an unmet need for more efficient therapies.

Among breast cancer subtypes, TNBC exhibits the highest lymphocyte infiltration, which was shown to positively correlate with better prognosis and treatment responses in several studies [5,6,7]. Therefore, patients with TNBC may benefit from cell-based immunotherapies. In particular, chimeric antigen receptor (CAR) T cells, which are genetically engineered to target cancer-specific antigens, have proven to be promising therapeutic options in various malignancies [8,9]. However, while this approach has demonstrated high efficacy in treating hematological malignancies, several obstacles hinder its clinical success in solid tumors so far. Among them, the occurrence of severe adverse effects due to a lack of suitable target antigens as well as cytokine release syndrome (CRS) due to excessive cytokine secretion have affected treatment safety [10,11]. Moreover, a lack of long-term in vivo persistence challenges durable remissions in the therapy of hematological malignancies [12,13] and has also been reported by several researchers in solid tumor treatment [14,15]. A number of factors are known to contribute to the decline in persistence, including CAR T cell exhaustion and differentiation towards an effector-like phenotype (extensively reviewed in [16]). Thus, various attempts have been made to fine-tune these effector functions via adjustments of the CAR structure [17].

In general, CARs are comprised of several domains, each of which has specialized effects on CAR T cell functionality: an antigen-binding domain and a hinge domain enabling binding of the tumor antigen; a transmembrane (TM) domain connecting the extra- and intracellular parts of the CAR; one or more costimulatory domains for extended stimulation; and a stimulatory domain providing the primary T cell activating stimulus [18]. Currently, the majority of CAR structures incorporate a CD3ζ stimulatory domain, which is responsible for T cell activation via phosphorylation of its three immunoreceptor tyrosine-based activation motifs (ITAMs, *cis*-signaling) [19]. However, recent research suggests that the high level of phosphorylation achieved by the three ITAMs might be redundant, leading to diminished in vivo persistence [20,21]. As an alternative mechanism of CAR T cell activation, several groups demonstrated the possibility of *trans*-signaling, wherein the CAR interacts with naturally expressed signaling molecules via its TM domain derived from, for instance, the FcεRI γ-chain [22,23].

Similarly, in an attempt to enhance CAR T cell persistence for TNBC treatment, we designed and investigated an alternative architecture called endogenous signaling molecule activating (ESMA) CAR. Its composition lacks the CD3ζ stimulatory domain, utilizing *trans*-signaling via TM domains derived from CD335, CD336, or CD64 instead, which were shown to interact with the endogenous signaling molecules CD3ζ, DAP12, and the FcR γ-chain [24,25,26]. Compared to a cognate second generation CAR, epidermal growth factor receptor (EGFR)-directed ESMA CAR T cells showed sustained killing of the TNBC cell line MDA-MB-231 at lower velocity. Furthermore, exhaustion marker expression was reduced and cytokine secretion was diminished. In an in vivo xenograft mouse model, the lead candidate CD335 ESMA CAR demonstrated promising anti-tumor efficacy, characterized by an enhanced memory-like phenotype and pronounced tumor infiltration. Taken together, these observations illustrate the potential of the ESMA CAR T cells for TNBC treatment.

## 2. Results

### 2.1. Activation of T Cells through the Interaction of CAR Transmembrane Domains with Endogenous Signaling Molecules Sufficiently Stimulates T Cell Killing Capacity

To test the potential of CARs to initiate T cell activation upon interaction of their TM domains with different endogenously expressed signaling molecules, several CARs were designed that incorporate different TM domains derived from CD335, CD336, and CD64 immune cell receptors but lack a CD3ζ domain (Figure 1a). Following lentiviral transduction, all ESMA CARs, which are directed against EGFR via a Cetuximab-derived scFv, were stably expressed on the T cell surface, varying between 25% to 50% in transduction efficiency for different T cell donors (Figure 1b). A cognate second generation CAR containing a CD8α TM domain and CD3ζ stimulatory domain used as a positive control (pos. Ctrl.) reached transduction efficiencies between 30% and 70%. Untransduced T cells (UnTd) were used as a negative control.

Subsequently, the CAR T cell killing capacity was established in various in vitro assays. For this, co-cultures with the EGFR-expressing MDA-MB-231 target cell line were set up. First, an imaging-based assay was used to assess killing kinetics over time (Figure 1c). Compared to the UnTd and target cell only negative controls, target cell growth was diminished by all tested CAR T cells. The ESMA CAR killing was delayed compared to the pos. Ctrl. and the CD335 CAR showed the strongest killing capacity, whereas CD336 and CD64 showed similar killing kinetics to one another. Additionally, serial target cell killing was assessed in a repetitive killing assay (Figure 1d). Here, new target cells were added every second day to assess the sustained killing capacity of the CAR T cells when they were exposed to target cells repetitively over an extended period of time. All ESMA CARs diminished target cell growth at comparable levels to the pos. Ctrl. and no declining killing capacity over time was detected. When comparing T cell counts, differences in T cell proliferation were observed over time. Whereas ESMA CAR T cells proliferated rapidly on the first days of co-culturing, the pos. Ctrl. showed a comparatively slower increase. At the end of the seven-day culture, similar CD335 and pos. Ctrl. CAR T cell counts were reached, whereas CD336 and CD64 ESMA CAR T cell counts were lower. In contrast, there was no proliferation of UnTd T cells and CAR T cells in the monoculture. Thus, T cell activation via the ESMA CAR can induce sustained killing of target cells over an extended period of time and after repetitive exposure to antigens.

### 2.2. T Cell Exhaustion and Cytokine Secretion Are Reduced in ESMA CARs

Besides cytotoxic capacity, additional effector functions of the ESMA CARs were analyzed. Cytokine secretion of the proinflammatory cytokines GM-CSF, IFNγ, TNFα, and IL-2 was assessed following an overnight co-culture of CAR T cells with MDA-MB-231 target cells (Figure 1e). Despite donor-dependent differences (n = 4) in the level of cytokine secretion, GM-CSF and IFNγ were secreted at the highest levels. Compared to the pos. Ctrl. CAR T cells, all ESMA CARs showed up to 50% diminished cytokine secretion, with CD335 CAR T cells secreting the highest cytokine levels out of the three tested CARs. Only minimal TNFα levels close to or below the detection limit were secreted by the ESMA CARs. In contrast to the other cytokines, IL-2 was elevated in T cell only controls and reduced in ESMA CAR co-cultures, whereas the pos. Ctrl. CAR T cells secreted high IL-2 levels that exceeded the assay’s upper detection limit. Overall, pro-inflammatory cytokine secretion by ESMA CAR T cells was upregulated specifically upon co-culture with target cells, compared to the pos. Ctrl. CAR T cells at markedly lower level.

In order to further characterize CAR T cells, exhaustion marker phenotyping (Lag-3, Tim-3, PD-1) was performed following co-cultures (Figure 1f). Merely minimal expression was detectable in CAR T cell only controls. Upon co-culturing, exhaustion marker expression was upregulated in all CAR T cells; however, the pos Ctrl. expressed up to 50% higher levels of Lag-3 and PD-1 compared to ESMA CAR. In contrast, Tim-3 expression levels were almost doubled in ESMA CAR T cells. Nevertheless, the percentage of ESMA CAR T cells expressing all three exhaustion markers—which was considered as a characteristic for terminally exhausted T cells—was reduced to one third compared to the pos. Ctrl., showing a lower overall exhaustion of ESMA CAR T cells in co-culture. Thus, the in vitro killing capacity of ESMA CAR T cells coincided with reduced cytokine secretion and diminished T cell exhaustion upon antigen stimulation.

### 2.3. CD335 ESMA CAR T Cells Diminish Tumor Growth in Xenograft In Vivo Model

To analyze how the observed in vitro characteristics of the ESMA CAR T cell effector function translate in vivo, a xenograft study with immunocompromised *NOD SCID Gamma (NSG)* mice was performed (Figure 2a). Mice bearing a subcutaneous MDA-MB-231 cell line-derived tumor were treated with CD335 ESMA CAR T cells, pos. Ctrl. CAR T cells, or UnTd T cells. The tumor burden was monitored longitudinally using fluorescence imaging (FLI) of the TurboRFP-expressing tumors (Figure 2b). From Day 7 onward, tumor growth in the pos. Ctrl. was significantly diminished compared to the UnTd control and reached the detection limit after approximately 14 days (Figure 2c). However, four mice of the pos. Ctrl. group had to be taken out before the end of the study as they reached the endpoint criteria (Figure 2d). On the other hand, tumors in the UnTd group continued growing until reaching a plateau phase after approximately 30 days. Compared to the UnTd control, tumor growth was significantly reduced in the CD335 ESMA CAR-treated group. There was a time delay compared to the pos. Ctrl., as tumor growth first stagnated in the CD335 ESMA CAR group and was approaching the detection limit by the end of the study in all mice except one (Appendix A). Overall, the study showed that CD335 ESMA CAR T cells could significantly control tumor growth over time, albeit at lower velocity than the pos Ctrl. CAR T cells.

### 2.4. ESMA CAR Design Delays T Cell Differentiation and Effector Function Kinetics In Vivo

To characterize the functional state of the CAR T cells throughout the in vivo study, blood was extracted from the facial veins of mice each week and subjected to T cell phenotypic analysis (Figure 3a). The percentage of human (h)CD3+ cells out of all leukocytes was used to compare T cell levels in blood over time. After an initial peak on Day 7, T cell levels in blood dropped down below 10%. Close to the endpoint, T cell levels started rising again in all groups. When evaluating T cell subtypes, it could be seen that UnTd T cell composition stayed stable until the study endpoint, with an observed ratio of around 50% CD4+ and CD8+ T cells. In contrast, CD8+ T cell percentage rapidly increased in the pos. Ctrl. group, then dropped again after tumor eradication at around Day 14. For the CD335 ESMA CAR group, a similar trend was visible; however, a slower but continuous shift towards CD8+ T cells was observed, which was linked to the slower tumor eradication. Thus, the phenotypic analysis of T cells in blood supported the in vivo killing kinetics, as CD335 ESMA CAR T cells continued to proliferate steadily over an extended period of time.

In addition to T cell phenotypic analysis, levels of human cytokines in the blood sera were quantified using the MACSPlex technique (Figure 3b). As overall cytokine secretion was close to or below the lower detection limit in some mice in the first two weeks, secretion levels could not be quantitatively compared between days. Corresponding to the observed cytokine concentrations in vitro, mainly GM-CSF and IFN-γ were secreted by the T cells while only low levels of IL-2 and almost no TNF-α cytokines were detected. Similarly to the phenotypic differences between the pos. Ctrl. and the CD335 ESMA CAR group, their cytokine secretion profiles were also distinct. Whereas cytokine secretion was elevated for the pos. Ctrl. on Day 7 and shortly dropped on Day 14 as tumors were eradicated, it steadily increased in the CD335 ESMA CAR group over time. Overall, the analysis showed durable cytokine secretion by the CD335 ESMA CAR cohort, with slightly lower GM-CSF and equal levels of IFN-γ compared to the pos. Ctrl. CAR group.

### 2.5. CD335 ESMA CAR T Cells Show Reduced Exhaustion and a Memory-like Phenotype

To apprehend the in vivo differentiation of ESMA CAR T cells, the expression of the memory markers CD45RA and CCR7 was analyzed in different organs at the study endpoint (Figure 3c). Their phenotype in different organs of one group was mostly uniform, with T cells in the bone marrow showing slightly lower differentiation. In all groups, effector memory (T_em_) cells had the highest abundance, and almost no naïve (T_n_) and central memory (T_cm_) cells were detected. The main differences between groups were in the percentages of T_em_ and terminally differentiated (T_emRA_) cells. CD335 ESMA CAR T cells showed a lower percentage of T_emRA_ cells compared to the pos. Ctrl. group and slightly increased T_em_ expression compared to the UnTd control group. This less-differentiated phenotype of CD335 ESMA CAR T cells contrasts with the observed phenotype pre-injection (Appendix A), where T_emRA_ cells were the most abundant cell type. Taken together, the CAR design affects T cell differentiation in vitro and in vivo. In the case of CD335 ESMA CAR T cells, a less differentiated, more memory-like phenotype was induced that resembled the UnTd group rather than the pos. Ctrl.

Additionally, the expression of exhaustion markers Lag-3, Tim-3, and PD-1 at the study endpoint was analyzed (Figure 3d). In contrast to in vitro, Lag-3 and PD-1 expression was upregulated in all groups, whereas only minimal Tim-3 expression was detected. When comparing the different organs, the expression of exhaustion markers was highest in bone marrow and lowest in spleen tissue. Overall, the UnTd group showed the highest exhaustion marker expression in each organ. Strikingly, strong differences between the expression levels in CD4+ and CD8+ T cell subtypes could be observed in each group. All three groups showed the same tendency for each T cell subtype; for instance, Lag-3 expression was doubled in CD8+ T cells and close to 100%. Similarly, Tim-3 expression was also upregulated in CD8+ T cells, albeit at a lower level, ranging between 10 and 40%. In contrast, PD-1 expression was lower in CD8+ compared to CD4+ T cells. Compared to the UnTd control, CD335 ESMA CAR T cells showed lower expression levels across exhaustion markers and organs. Additionally, Lag-3 expression was significantly reduced compared to the pos Ctrl. group in some organs. Taken together, it could be shown that exhaustion marker expression is reduced in CD335 ESMA CAR T cells and that there are strong T cell subtype-specific differences in their expression levels.

### 2.6. ESMA CAR T Cells Exhibit Robust Tumor Infiltration

To complement the phenotypic analysis in immune organs, the spatial distribution of T cells within the tumor was analyzed using a cyclic immunofluorescent (IF) imaging approach (Figure 4a). In contrast to the UnTd control, where no T cells could be detected, there was a strong T cell infiltration into the CD335 ESMA CAR-treated tumor. To quantify the T cell distribution and penetration into the tumor core, the tumor was divided into three regions (Appendix A): an inner core region, an outer region, and a distinct border region characterized by high smooth muscle actin (SMA) expression (Figure 4b). The majority of CD4+ and CD8+ T cells were located in the outer tumor region, with lower percentages in the border and inner regions. The majority of tumor cells, characterized by their ErbB2 expression, were located in the inner and border region. In line with the blood analysis, CD8+ T cells were more abundant than CD4+ T cells. They also showed enhanced tumor infiltration, as higher percentages of CD8+ T cells were localized in the border and core regions.

To further characterize the tumor-infiltrating T cells, different activation and exhaustion markers were stained (Figure 4c). CD4+ T cells especially expressed CD134, showing their activation at the tumor site both at the edge of and within the tumor. In line with the blood analysis, there were subtype-specific differences in exhaustion marker expression, as CD4+ T cells expressed higher PD-1 levels whereas Lag-3 was abundant on both subtypes. Moreover, expression levels also differed between tumor regions, as T cells in the outer and border regions showed higher expression of all activation and exhaustion markers independent of the subtype.

To assess the interactions between T cells and tumor cells, the CD335 ESMA CAR-treated tumor was characterized and compared to the UnTd control (Figure 5a). In the CD335 ESMA CAR-treated tumor core and border, a more pronounced cytoskeletal structure characterized by higher cytokeratin 7 (CK7) and SMA expression compared to the outer tumor region could be observed (Figure 5b). Combined with the increased expression of the immune inhibitory marker PD-L1, the inner and border region have a denser structure than the outer region. When comparing the UnTd treated tumor, less pronounced differences in PD-L1, CK7, and SMA expression between the tumor regions could be observed, showing a more homogenous tumor structure with a less distinct border zone (Figure 5c). Expression of the cell death marker p53 was similarly enhanced in the tumor core for both treatment groups. Together, the T cell and tumor marker staining illustrated strong CD335 ESMA CAR T cell infiltration into the tumor, suggesting a disruption of the tumor architecture from the periphery towards the core. Thus, the anti-tumor efficacy of the ESMA CAR T cells against the TNBC cell line MDA-MB-231 could be verified by different means.

## 3. Discussion

In this study, a modified CAR T cell architecture was designed and evaluated with the aim of increasing the limited CAR T cell in vivo persistence, which is commonly challenging in solid tumor treatments [17,27,28]. The ESMA CAR T cells exerted anti-tumor effects both in vitro and in vivo while simultaneously displaying differing effector function kinetics compared to a cognate second generation CAR.

Initially, the majority of CARs incorporate a CD3ζ stimulatory domain that naturally contains three ITAMs. On top of that, it has been shown that increased ITAM multiplicity in CARs enhances receptor potency as a higher number of T cells become activated [29]. However, the level of activation achieved by all three CD3ζ ITAM subunits can be excessive and limit T cell persistence [20,21]. In a study by Feucht et al., CARs with two mutated CD3ζ ITAMs outperformed conventional three ITAM-containing CARs in a xenograft in vivo study concomitantly with enhanced in vivo persistence and a shift towards less-differentiated T cell subtypes [20]. Notably, several studies have illustrated that there is potential in employing endogenous cell components to tweak CAR functionality. Early research on chimeric receptors already illustrated that CARs with special domains derived from, for instance, TCR α- and β-chain [8] or FcεRI γ-chain [22] could associate with endogenous signaling molecules such as CD3ζ. Additionally, a study by Bridgeman et al. determined that CD3ζ-based CARs led to the formation of heterodimers with endogenous CD3ζ via its TM domain [23]. This so-called *trans*-signaling then led to T cell activation even if all CAR ITAMs were mutated. In an effort to mimic the natural TCR signaling cascade with its division into separate stimulatory and costimulatory molecules, CARs comprising multiple co-transduced receptor chains that interact with one another have been created before [30,31,32]. While those multi-chain CARs could induce potent anti-tumor effects both in vitro and in vivo, the co-transduction of multiple constructs comes with additional challenges. T cell transduction via bicistronic lentiviral vectors challenges vector packaging capacity whereas transduction of multiple vectors reduces the number of double positive cells, which both adversely affect transduction efficiency [33,34]. Thus, in this study we chose to forego a multichain approach, instead relying on endogenously expressed receptors for T cell stimulation in combination with the CAR.

TM domains derived from immune cell receptors CD335, CD336, and CD64, which are known to interact with different ITAM-containing receptors, were incorporated into the CAR [24,25,26,35]. In the ESMA CARs, T cell activation is solely reliant on the interaction of CAR and receptor TM domains in an effort to avoid the aforementioned redundant *cis*-signaling by multiple CD3ζ ITAMs and improve persistence. Notably, even though both CD335 ESMA CAR and cognate second generation CAR use CD3ζ for T cell stimulation, the *cis-* and *trans*-signaling seem to introduce differences, leading to the observed variance in effector functions. Therefore, these signaling mechanisms appear to not be directly interchangeable. One possible explanation is the rapid downregulation of the TCR complex including CD3ζ, which prevents overstimulation, as it is observed upon TCR-mediated T cell activation [36,37]. In CAR T cell therapy, CAR regulation upon activation via, for instance, targeted integration into differentially regulated gene loci led to promising in vitro and in vivo performance [38,39].

In clinical settings, robust initial T cell expansion following infusion as well as long-term persistence have both been linked to better clinical outcomes and durable remission [12,13,28,40,41,42], illustrating the incentive to enhance T cell persistence by modulating CAR signaling. The influence of the CAR design on CAR T cell phenotype and effector function, especially the effect of CAR costimulatory domains CD28 and 4-1BB, has been studied in detail. Whereas both induced similar downstream phosphorylation of proteins, CD28 induced more intensive and rapid signaling [17,27]. However, this did not improve the in vivo efficacy of the CD28 CAR T cells. In contrast, a slower velocity of T cell activation and cytotoxicity might prove beneficial for long-term effector function, as CD28 CARs showed increased exhaustion, reduced tumor elimination, and limited persistence in clinical settings compared to 4-1BB CARs [27,43,44,45,46]. Notably, a lower level of exhaustion and reduced cytokine secretion have also been observed in T cells with engineered TCR (TCR-T cells), which showed similar dose responses to a cognate CAR [47,48]. Excessive cytokine secretion has been linked to severe CRS in multiple CAR T cell therapies [49], adding increased patient safety to the incentives for CAR remodeling. Thus, it has been illustrated numerous times that multiple factors affect T cell potency besides CAR or TCR signaling strength, so a better balance between signaling intensity, durability, and safety might benefit overall therapy outcome.

To achieve this balance, we designed the ESMA CAR structure, expecting reduced stimulatory signaling, which should induce delayed yet durable anti-tumor effects. Indeed, this could be observed in the CD335 ESMA CAR in vivo study and further corroborated by the strong tumor infiltration visualized via immunofluorescent tumor imaging. Moreover, ex vivo phenotyping of the CD335 ESMA CAR T cells revealed elevated fractions of less-differentiated, memory-like T cells that have been shown to be beneficial for persistence and performance in clinical settings [28]. While this phenotype is indicative of extended persistence, it is not yet a direct predictor of extended effector function. Going forward, it should thus be further assessed whether ESMA CAR T cells retain their effector function, for instance, in a tumor rechallenge model similar to the one described by Tomar et al. [50]. Combined with the data gathered here, an extended in vivo study would further increase the predictive value of ESMA CAR functionality in a clinical setting.

Furthermore, the in vivo functionality as well as differential gene expression related to CAR signaling should also be investigated for CD336 and CD64 ESMA CARs, as their in vitro performance differed from that of CD335 ESMA CAR, presumably due to differences in downstream signaling. Both CARs showed promising in vitro results characterized by the extended killing of target cells and reduced expression of exhaustion markers; however, the killing velocity and proliferative capacity were reduced compared to CD335 ESMA CAR. In contrast to the abundantly expressed CD3ζ [51], expression of the CD336-interacting receptor DAP12 as well as the CD64 counterpart FcR-γ chain is limited to different T cell subtypes [52,53,54,55]. Additionally, DAP12 [56] and the FcR-γ chain [57] each contain one ITAM whereas CD3ζ contains three ITAMs per receptor chain [58]. Thus, the T cell subtypes that can be activated by the CARs are more limited in CD336 and CD64 ESMA CARs. As the in vivo study illustrated the potential of CD335 ESMA CAR T cells as a treatment against the TNBC cell line MDA-MB-231, the focus should be broadened towards other TNBC cell lines, other target antigens apart from EGFR, and other indications besides TNBC. In this context, CD336 and CD64 ESMA CARs should be carefully investigated; for instance, their potential in targeting high-level EGFR-expressing tumors should be evaluated. As a follow up, a dual targeting approach against two TNBC antigens, similar to the tandem CD19/CD20 CAR described by Shah et al., could then be used to further increase treatment efficacy and safety in different settings [59].

The expression of EGFR, as well as most solid tumor antigens, is not restricted to malignant cells only; thus, multiple CAR T cell approaches have been tested with the aim to limit the on-target, off-tumor toxicities of for instance EGFR-targeting CAR T cells and increase their safety [60,61]. While these studies focused on tuning CAR sensitivity via lower affinity scFvs, we chose to use a high-affinity Cetuximab-derived scFv and reduced CAR signaling strength via the endogenous signaling approach instead. As a result, high-level EGFR expression will likely be necessary to induce anti-tumor effects. The ESMA CAR approach is therefore aimed at antigens that are overexpressed in tumors and whose treatment has so far been challenged by background expression in healthy tissues. Moreover, as the ESMA CAR-interacting endogenous receptors are also expressed on NK cells [54,62], their applicability in a CAR NK cell setting should be evaluated, especially considering recent promising in vivo studies of EGFR-directed CAR NK cells against glioblastoma and breast cancer brain metastasis [63,64]. This way, the potential of the newly established ESMA CAR approach could be further elucidated in different settings.

Whereas CAR and TCR activation are usually not interconnected but instead induce additive effector functions [65], ESMA CARs bridge the gap and employ synergistic mechanisms of both CAR costimulation and (TCR) endogenous stimulatory signaling. This way, redundant signaling of multiple CAR ITAMs that could induce early T cell anergy and imposes different safety risks could be avoided. Instead, the T cell phenotype is directed towards a memory-like state with overall reduced T cell exhaustion and cytokine secretion levels both in vitro and in vivo. Going forward, their applicability as treatment options against different (TNBC) target antigens should be further evaluated both in T and NK cell settings, with a focus on highly expressed tumor antigens that are challenged by on-target, off-tumor toxicities, CRS, or limited in vivo persistence.

## 4. Materials and Methods

### 4.1. Transgene Construction

All chimeric antigen receptor (CAR) constructs were designed to contain a single-chain variable fragment (scFv) as the antigen binding domain, which was derived from the clinical monoclonal antibody Cetuximab [66]. Positive control (pos. Ctrl.) CARs were composed in a second generation CAR structure, containing a CD8α hinge and transmembrane domain, connected to a 4-1BB co-stimulatory and a CD3ζ stimulatory domain. Endogenous signaling molecule activating (ESMA) CARs consisted of an scFv connected to a CD8α hinge, a transmembrane (TM), and a 4-1BB costimulatory domain lacking a CD3ζ intracellular domain. ESMA CAR constructs differed based on their TM domain (derived from TM domains of CD335, CD336, or CD64). Sequences of the aforementioned human protein domains were taken from UniProt, and codon optimization for expression in human cells was performed using ATUM’s gene synthesis services. In silico gene assembly was conducted using CloneManager software (Version 9), while plasmids were cloned using restriction digestion and amplification in NEB^®^ 5-alpha-competent *E. coli* (NEB).

### 4.2. Lentiviral Vector Production

To generate VSV-G pseudotyped lentiviral particles encoding the CAR of interest, lentiviral vector production was conducted as previously described by Schäfer et al. [67]. Briefly, HEK293T (ATCC, Manassas, VA, USA) cells were cultured in Dulbecco’s Modified Eagle’s medium (DMEM, Biowest, Nuaillé, France) supplemented with 10% fetal bovine serum (FBS, Catus Biotech, Tutzing, Germany) and 100 µg/mL Primocin (InvivoGen, San Diego, CA, USA) to a confluency of 70–90%. Transfection was performed with a 1:2 DNA:MACSfectin™ (Miltenyi Biotec, Bergisch Gladbach, Germany) mixture with a total amount of 50 µg DNA per 175 cm^2^ flask. Sodium butyrate (Merck, Darmstadt, Germany) was added 20 h after transfection to a final concentration of 10 mM, and the lentiviral supernatant was harvested 48 h after transfection. Following a spinoculation for 16 to 20 h at 3500× *g* and 4 °C, virion-containing pellets were resuspended at a 200-fold concentration in TexMACS™ Medium (Miltenyi Biotec) and stored at −80 °C until further usage.

### 4.3. CAR T Cell Generation

Buffy coats from healthy anonymous donors were obtained from the German Red Cross Dortmund or Hagen as registered and approved by the Ethics Committee of the German Red Cross. CAR T cell generation was performed as described previously by Kinkhabwala et al. [68]. In brief, peripheral blood mononuclear cells (PBMCs) were isolated using density gradient centrifugation. T cell enrichment was conducted using the human Pan T Cell Isolation Kit (Miltenyi Biotec) according to the manufacturer’s protocol. T cells were cultivated in TexMACS™ Medium (Miltenyi Biotec) containing 200 IU/mL human interleukin-2 (IL-2, research grade, Miltenyi Biotec) and 100 µg/mL Primocin (InvivoGen) (expansion medium) and activated with human TransAct™ reagent (Miltenyi Biotec). Twenty-four hours after T cell isolation, activated T cells were transduced with 40–50 µL of CAR-encoding lentiviral vector. Forty-eight hours after T cell transduction, the TransAct™ and residual lentiviral vector were removed through an exchange with fresh expansion medium. T cells were expanded for 12 to 15 days in total, and CAR expression levels were analyzed on Day 7 and Day 12 or 14 of expansion through flow cytometry. After expansion, T cells were used for in vitro functionality testing or subjected to in vivo studies.

### 4.4. In Vitro Analysis

#### 4.4.1. Killing Assay

CAR T cell-mediated cytotoxicity was measured through a life-cell imaging killing assay using the IncuCyte^®^ S3 system (Sartorius, Göttingen, Germany). For the co-cultures, 2 × 10^4^ GFP-expressing target cells (MDA-MD-231, ATCC) were seeded in a flat-bottomed 96-well plate, allowed to settle down, and incubated overnight under standard cultivation conditions (37 °C, 5% CO_2_, 95% humidity) in DMEM (Biowest) supplemented with 10% FBS (Catus Biotech) and 100 µg/mL Primocin (InvivoGen). Afterwards, CAR T cells were added to the target cells at an effector:target cell (E:T) ratio of 2:1. The co-culture plate was introduced into the plate reader and four images per well were taken every 2 h for a duration of at least 4 days. Using the IncuCyte^®^ S3 analysis software (Sartorius, Version 2019A), the area and integrated intensity of the GFP-expressing target cells over time was analyzed.

#### 4.4.2. Co-Culture for In Vitro Phenotyping

To analyze differential marker expression and cytokine secretion upon co-culturing with target-expressing cells, 1 × 10^5^ GFP-expressing MDA-MB-231 cells and 5 × 10^4^ CAR T cells were co-cultivated in a round-bottomed 96-well plate under standard cultivation conditions (37 °C, 5% CO_2_, 95% humidity) in DMEM (Biowest) supplemented with 10% FBS (Catus Biotech) and 100 µg/mL Primocin (InvivoGen). After 20 to 24 h, the supernatant was used for a cytokine assay and CAR T cells were stained for exhaustion markers as described below.

### 4.5. Flow Cytometry

The cellular composition of isolated T cells was analyzed with human CD3 (hCD3, REA613), hCD4 (REA623), and hCD8 (REA734) monoclonal recombinant antibodies (Miltenyi Biotec) according to manufacturer’s staining protocol. To determine the transduction efficiencies and analyze the CAR expression profiles, respectively, T cells were identified using hCD3 (REA613) monoclonal recombinant antibody (Miltenyi Biotec) and CAR expression was analyzed using a His-tag-labeled recombinant human EGFR protein (10 min incubation at 4 °C, 1:20 dilution, Acrobiosystems, Newark, DE, USA), which was counterstained with monoclonal anti-His antibody (GG11-8F3.5.1, Miltenyi Biotec) according to manufacturer’s protocol. Cell viability was assessed using propidium iodide (Miltenyi Biotec). Stained samples were acquired using a MACSQuant^®^ Analyzer 16 (Miltenyi Biotec) and data was analyzed using MACSQuantify™ software 2.13 (Miltenyi Biotec).

#### 4.5.1. Marker Staining

As previously described, cells were stained for different markers in order to assess cellular phenotypes in vitro, during the in vivo study (blood), and ex vivo at the study endpoint (blood, spleen, bone marrow). To identify human T cells in the blood and organs of mice over the course of the in vivo study, murine CD45 (mCD45, REA737) and mTer-119 (REA847) monoclonal recombinant antibodies (Miltenyi Biotec) were used in combination with hCD3 (REA613) or hCD4 (REA623) and hCD8 (REA734). Aside from general T cell characterization with hCD3, hCD4, and hCD8, different marker panels were used for respective analysis. The exhaustion phenotype was evaluated using hCD223 (Lag-3, REA351), hCD279 (PD-1, REA1165), and hCD366 (Tim-3, REA635) antibody (Miltenyi Biotec). The staining of memory markers was performed using hCD197 (CCR7, REA546) and hCD45RA antibodies (REA562, Miltenyi Biotec) according to the manufacturer’s protocol.

#### 4.5.2. Repetitive Killing Assay

The serial killing capacity of CAR T cells was evaluated in a repetitive killing assay, in which effector cells were rechallenged with new target cells every second day; a modified version of the repetitive killing assay described by Wang et al. [69] was used. Four identical plates were prepared in triplicates as follows:

GFP-expressing MDA-MB-231 cells were prepared by detaching them from the cell culture flask with cold PBS/EDTA/BSA (PEB, autoMACS^®^ rinsing solution supplemented 1:20 with MACS^®^ BSA stock solution, Miltenyi Biotec) to avoid digestion of surface proteins. The target cell density was adjusted to 1.6 × 10^5^ cells/mL, then 100 µL of cells was plated in a round-bottomed 96-well plate. The effector cell density was adjusted to 3.2 × 10^5^ CAR+ T cells/mL, then 100 µL per well was added to reach an E:T ratio of 2:1. Cells were incubated under standard cultivation conditions (37 °C, 5% CO_2_, 95% humidity) in 200 µL DMEM (Biowest) supplemented with 10% FBS (Catus Biotech) and 100 µg/mL Primocin (InvivoGen).

On Day 1, 3, 5, and 7, one plate was taken out for analysis and the cells were detached using cold PEB. Then, cells were counted using a MACSQuant^®^ Analyzer 16 (Miltenyi Biotec) and MACSQuantify™ analysis software 2.13 (Miltenyi Biotec). The remaining plates were further cultivated and rechallenged with fresh target cells on Day 2, 4, and 6. For the rechallenge, 100 µL of supernatant was removed and 3.2 × 10^4^ GFP-expressing target cells in 100 µL of medium were added.

#### 4.5.3. Cytokine Assay

To determine the levels of human cytokines in the supernatants of in vitro CAR T assays and blood sera of mice, the human MACSPlex Cytokine 12 Kit (Miltenyi Biotec) was used according to the manufacturers’ instructions. For the in vitro assays, 50 µL of the supernatant of the co-cultures for phenotyping was used. During the in vivo study, blood was extracted every seven days, and 1:2 or 1:3 dilutions of the blood sera were conducted. The concentrations of the following cytokines were evaluated: GM-CSF, IFN-α, IFN-γ, lL-2, IL-4, IL-5, IL-6, IL-9, IL-10, IL-12, IL-17A, and TNF-α; these were measured using a MACSQuant^®^ Analyzer 10 (Miltenyi Biotec) and analyzed using the MACSPlex InspectoR web app, Version 0.7.1 (Miltenyi Biotec).

#### 4.5.4. Data Analysis with the MACSPlex InspectoR Web App

The automated data analysis of measurements obtained with the MACSPlex Cytokine 12 Kit for humans was performed with the in-house developed software, the MACSPlex InspectoR Shiny app [70] (see Appendix A for a detailed description). First, single bead events were selected on the FSC-A and SSC-A channels. The different analyte bead populations were then identified through clustering and assignment of the analytes by aligning the median intensities to a grid [71]. A standard curve was calculated after local Hermite fitting with linear interpolation and used to calculate the concentrations of the different analytes in the samples. The downloaded results tables of the concentrations were used for comparative analysis in downstream applications.

### 4.6. In Vivo Analysis

#### 4.6.1. Tumor Mouse Model

All experiments were performed according to guidelines and regulations and were approved by the Governmental Review Committee on Animal Care in North Rhine Westphalia (NRW), Germany (Landesamt für Natur, Umwelt and Verbraucherschutz NRW, Approval number 84-02.04.2017.A021). Tumors were engrafted in *NOD SCID Gamma (NSG; NOD.Cg-PrkdcscidIl2rgtm1Wjl/SzJ)* mice (Jackson Laboratory, Bar Harbor, ME, USA, provided by Charles River). To do so, the triple-negative breast cancer (TNBC) cell line MDA-MB-231 was genetically modified to express the fluorescent reporter protein TurboRFP and underwent single-cell clone expansion cultivated in DMEM with 10% FBS (Catus Biotech). For tumor establishment, 1 × 10^6^ MDA-MB-231 TurboRFP cells were injected subcutaneously (s.c.) in 100 µL of Dulbecco’s phosphate-buffered saline (DPBS, *w*/*o* Ca^2+^ and Mg^2+^, VWR, Darmstadt, Germany) in the right flank of NSG mice. Tumor growth was monitored using caliper measurements and fluorescence imaging (FLI) over time. Seven days after tumor engraftment, mice were randomized based on the fluorescent intensity of the tumor and the respective tumor size, resulting in evenly distributed groups with comparable levels of tumor burden. A total amount of 2 × 10^6^ CAR T cells (with a transduction efficiency of 33%) was injected into the tail vein of each animal in 100 µL of DPBS. Untransduced (UnTd) T cells served as a negative control and were adjusted to the total T cell numbers injected in the CAR groups (5.8 × 10^6^ cells). Over the course of the study, the well-being of mice was monitored and scored according to animal care guidelines. Mice were euthanized upon reaching humane endpoint criteria according to the guidelines or at the study’s end, and tumors were extracted for further ex vivo analysis.

#### 4.6.2. In Vivo Fluorescence Imaging

The tumor burden was monitored two to three times a week using FLI. Images were obtained using the optical in vivo imaging system (IVIS^®^) Lumina III (Perkin Elmer, Waltham, MA, USA). Briefly, mice were anesthetized using Vetflurane (Virbac, Carros, France) and placed into the imaging chamber, where images were acquired with an exposure time of 1 s and a filter combination of 560 nm for excitation and 620 nm for emission. Data was analyzed and quantified with the Living Image software 4.7.3 (Perkin Elmer). For quantitative analysis, regions of interest (ROIs) were set around the solid tumor area and values of average fluorescent radiant efficiency [$\frac{p/sec/{cm}^{2}/sr}{µW/{cm}^{2}}$] were determined and plotted over time to compare CAR T cell efficiencies.

### 4.7. Ex Vivo Analysis

Over the course of the in vivo study, 80 µL of blood was collected on Day 7, 14, 21, and 28 and at endpoint from the facial veins of the mice. Spleen tissue was collected from each mouse and dissociated using a gentleMACS Dissociator (Miltenyi Biotec) in Roswell Park Memorial Institute (RPMI, Biowest) medium, then pipetted through a 70 µm pre-separation filter (Miltenyi Biotec). Bone marrow was collected from each mouse by flushing the bone with RPMI using a syringe, followed by pipetting through a 70 µm pre-separation filter. Red blood cell lysis was performed with red blood cell lysis solution (Miltenyi Biotec), then cells were washed with PEB (Miltenyi Biotec) and blocked using PEB with 10% human and mouse FcR-blocking reagent (Miltenyi Biotec). T cells were characterized based on CAR expression and further phenotyped using different marker panels as previously described.

### 4.8. Cyclic Immunofluorescence Staining

Immunofluorescent (IF) stainings were performed using the MACSima™ Imaging Platform (Miltenyi Biotec). Xenografts were excised on Day 28 post-CAR T cell injection and incubated in 10% formalin, then neutral buffered (Sigma-Aldrich, St. Louis, MO, USA) overnight for formalin-fixed paraffin-embedded (FFPE) fixation. The next day, the tumors were washed in ddH_2_O for 30 min, followed by an ethanol dehydration series (70% to 100%) and two incubation steps in xylene isomer (Carl Roth, Karlsruhe, Germany) for 1 h. Tumors were then incubated in paraffin (Histosec^®^ without DMSO, Sigma-Aldrich) at 60 °C overnight, followed by embedding in paraffin molds.

After FFPE treatment, 3 µm tumor sections were cut on a Epredia™ HM 355S microtome (Thermo Fisher Scientific, Waltham, MA, USA), mounted on Epredia™ SuperFrost Plus™ slides (Thermo Fisher Scientific), and dried overnight at 40 °C. For deparaffinization, slides were incubated for 20 min in xylene, then a few seconds each in an ethanol series (100% to 50%), followed by storage in ddH_2_O until use. For antigen retrieval, slides were transferred to pH 9 TEC buffer (0.25 g Trizma base, 0.575 g EDTA disodium salt dihydrate, 0.32 g sodium citrate tribasic dihydrate, diluted in 1 L ddH_2_O and adjusted to pH 9 with NaOH). Slides were incubated in TEC buffer at 98 °C for 20 min, then inserted into MACSwell™ Four Imaging Frames (Miltenyi Biotec). DAPI pre-staining was performed, followed by introduction of the slides to the MACSima™ system, as described by Kinkhabwala et al. [68]. A ready-to-use internal prototype version of the REAscreen™ Immuno-oncology Kit (human, FFPE, version 01, Miltenyi Biotec), comprising a total of 61 fluorochrome-labeled antibodies, was used for sample characterization in 34 staining cycles. Data processing and stitching was conducted using MACS iQ View 1.1.0 (Miltenyi Biotec).

### 4.9. Statistics

Experimental data was analyzed using GraphPad Prism 9 (Graph-Pad Software, Boston, MA, USA). The analysis of the tumor burden was compared using one-way ANOVA. Statistical outcomes of in vivo experiments were organized in a pairwise significance matrix (PSM, Appendix A), where a comparison of two groups is represented by one box [72]. Significance (*p* ≤ 0.05) was indicated by a grey box. If one group had less than three animals left, no analysis for that group was performed, which is indicated by a black box. For the frequency analysis of different markers, two-way ANOVA was used. Statistical significance was illustrated by PSM as mentioned or by asterisks, (* *p* ≤ 0.05, ** *p* ≤ 0.01, *** *p* ≤ 0.001, **** *p* ≤ 0.0001).

## Figures and Tables

**Figure 1 ijms-25-00615-f001:**
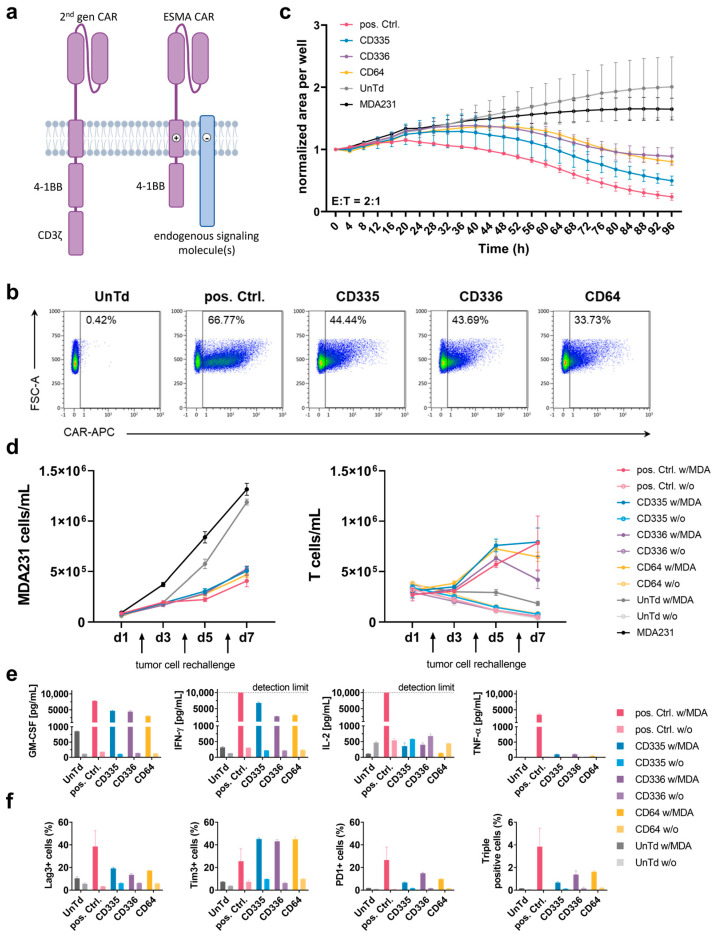
In vitro characterization of T cells expressing an endogenous signaling molecule activating chimeric antigen receptor (ESMA CAR) compared to a conventional second generation CAR. (**a**) Scheme and functional principle of a second generation CAR (pos. Ctrl.) and an ESMA CAR. In the second generation CAR, the CD3ζ stimulatory domain facilitates T cell activation. The ESMA CAR lacks a stimulatory domain; instead, T cell activation is achieved through interaction of the CAR transmembrane (TM) domain with the TM domains of endogenous signaling molecules. (**b**) Surface expression of epidermal growth factor receptor (EGFR)-directed pos. Ctrl. CAR and ESMA CARs (named after their respective TM domains) analyzed through flow cytometry. (**c**) Kinetics of CAR T cell cytotoxicity against green fluorescent protein (eGFP)-expressing MDA-MB-231 cells, detected using a fluorescence intensity-based timelapse imaging approach (IncuCyte S3 analyzer). The target cell area was normalized to the starting time depicted. (**d**) ESMA CAR T cells show sustained target cell killing and proliferative capacity in a repetitive killing assay measured through flow cytometry (n = 2). (**e**) ESMA CAR T cells display diminished cytokine secretion compared to the pos. Ctrl., measured following an in vitro 24 h effector-target cell co-culture. Cytokine levels quantified via flow cytometric MACSPlex assay and MACSPlex InspectoR web app (Appendix A). (**f**) Reduced expression levels of the following exhaustion markers were detected in ESMA CAR compared to the pos. Ctrl. CAR T cells following a 24 h co-culture: Lag-3, Tim-3, PD-1. (**c**–**f**) Quantitative analyses performed in triplicate with the mean ± standard deviation (SD) shown. Representative data from n ≥ 3 unless indicated otherwise.

**Figure 2 ijms-25-00615-f002:**
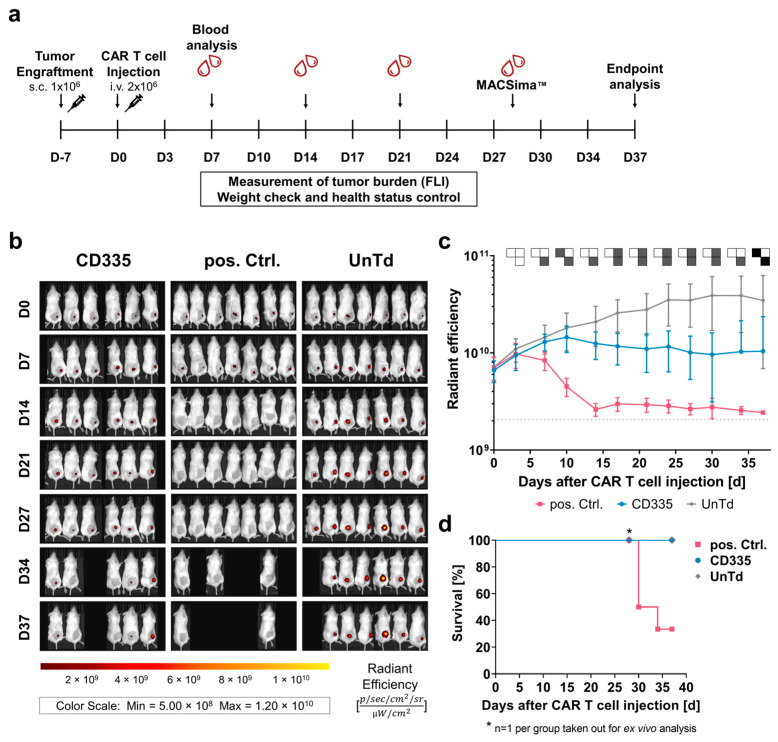
T cells expressing CD335 ESMA CAR display tumor burden control in vivo. (**a**) Schematic representation of the in vivo study workflow. Mice bearing a TurboRFP-expressing MDA-MB-231 cell line-derived xenograft solid tumor were randomized in three groups (n = 7 pos. Ctrl. & UnTd, n = 6 CD335) receiving CAR T cells expressing a conventional second generation CAR (pos. Ctrl), an ESMA CAR (CD335), or no CAR (UnTd). The tumor burden was monitored longitudinally by means of in vivo fluorescence imaging (FLI). Blood was analyzed weekly, ex vivo analyses of tumors (immunofluorescence) and immune organs (flow cytometry) were executed as indicated on either Day 28 or at the study end point, respectively. (**b**) Weekly depiction of CAR treated cohorts over the course of the study. (**c**) CD335 ESMA-CARs display tumoricidal efficacy with steady but slower kinetics when compared to the pos. Ctrl. over time. The UnTd control fails to diminish tumor growth. Detection limit at around 2.06 × 10^9^ (gray dotted line). Pairwise significance matrix (PSM, Appendix A) *p* ≤ 0.05 (grey), *p* > 0.05 (white), not assessable as n < 3 for one group (black) [one-way ANOVA, multiple comparisons]. (**d**) Kaplan–Meier curves indicate that in the pos. Ctrl. CAR T cell-treated mice, humane endpoint criteria are reached faster compared to the UnTd control and ESMA CAR T cell-treated cohorts.

**Figure 3 ijms-25-00615-f003:**
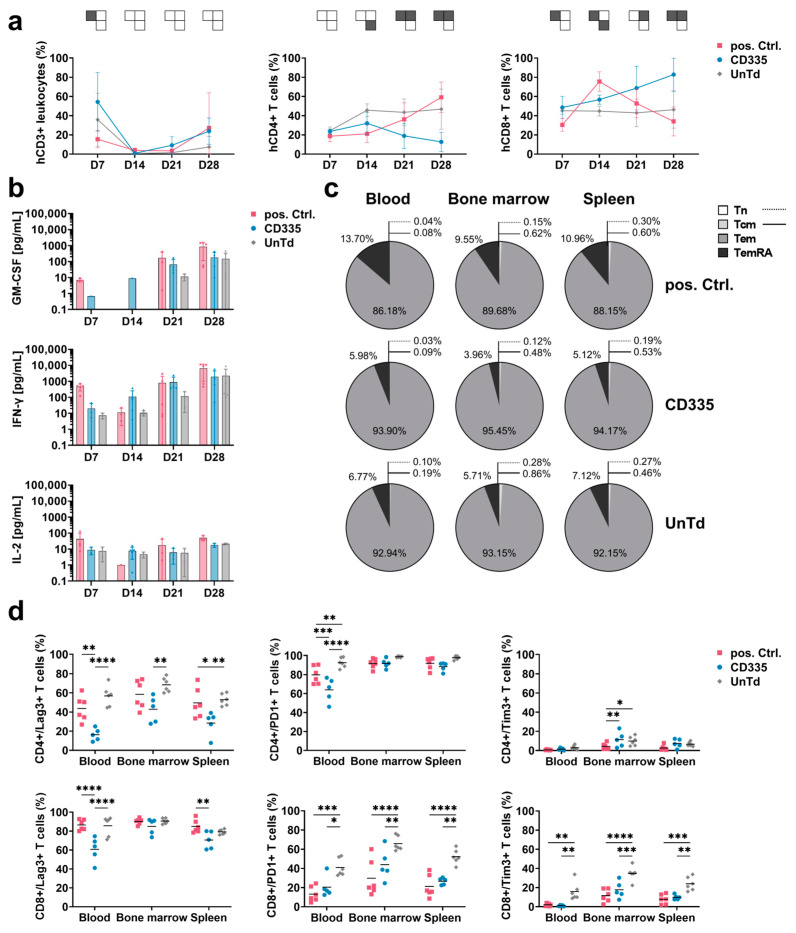
Ex vivo T cell phenotyping reveals differing kinetics and phenotypes of T cells expressing CD335 ESMA CARs vs. conventional second generation CARs. (**a**) Weekly analysis of T cell subtypes in blood sera shows diverging proliferation of CAR T cell subtypes over the course of the in vivo study. hCD3+ leukocytes were identified as a percentage of viable mTer119−/mCD45− single cells, then percentages of hCD4+ and hCD8+ T cells out of the hCD3+ cells were gated. Statistical analysis with two-way ANOVA (multiple comparisons); results arranged in PSM *p* ≤ 0.05 (grey), *p* > 0.05 (white). (**b**) Secretion of human pro-inflammatory cytokines in blood over time shows substantial cytokine secretion in the CD335 ESMA CAR-treated cohort, with slightly lower levels of GM-CSF and equal levels of IFN-γ compared to the pos. Ctrl. Cytokine levels quantified via flow cytometric MACSPlex assay and MACSPlex InspectoR web app (Appendix A). The mean ± SD and individual values are depicted. (**c**,**d**) Comparison of marker expression on T cells extracted at respective endpoints as measured through flow cytometry. In the pos. Ctrl. group, three mice reached the endpoint on Day 30 and one mouse on Day 34 of the study, while all other animals were taken out on Day 37. Human T cells were identified based on hCD4+ and/or hCD8+ expression. (**c**) Classification into T cell memory subtypes according to CD45RA and CCR7 surface expression. Percentages of naïve (Tn, CD45RA+CCR7+), central memory (Tcm, CD45RA−/CCR7+), effector memory (Tem, CD45RA−/CCR7−), and terminally differentiated effector memory T cells (TemRA, CD45RA+/CCR7−) are depicted. (**d**) Comparison of exhaustion marker (Lag-3, Tim-3, and PD-1) expression on hCD4+ and hCD8+ T cells, respectively. Quantification reveals reduced exhaustion marker expression in the CD335 ESMA CAR group with T cell subtype dependent variances. Statistical analysis with two-way ANOVA (multiple comparisons), * *p* ≤ 0.05, ** *p* ≤ 0.01, *** *p* ≤ 0.001, **** *p* ≤ 0.0001.

**Figure 4 ijms-25-00615-f004:**
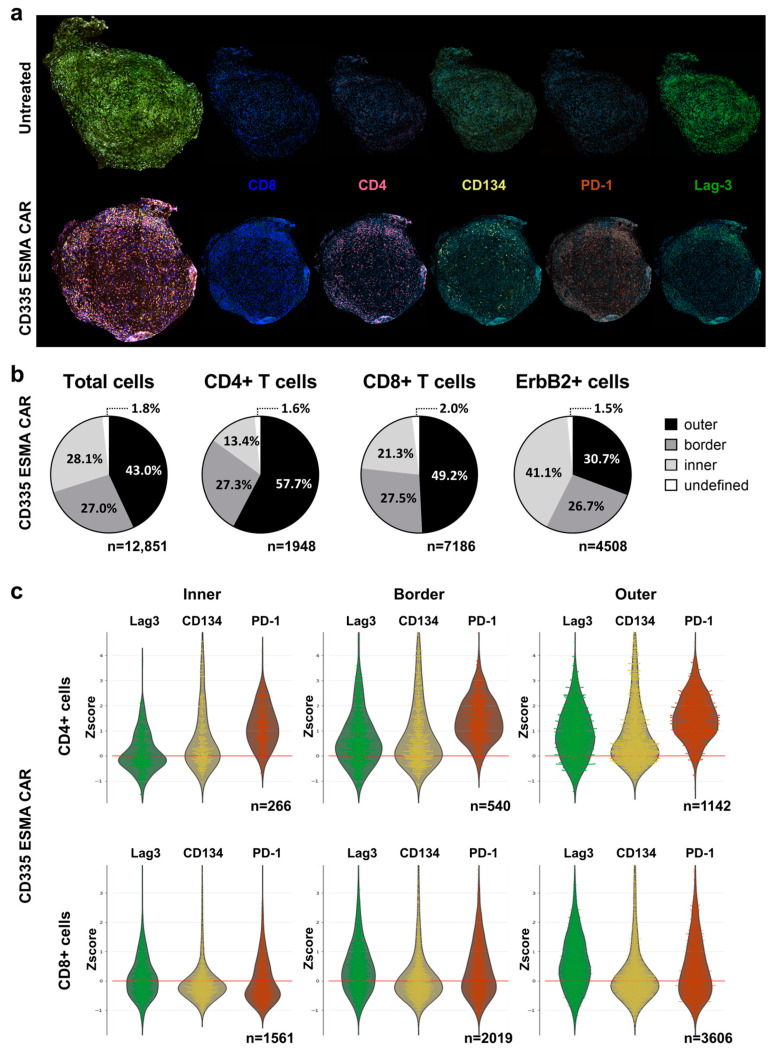
Analysis of intra-tumoral spatial T cell distribution using the cyclic immunofluorescent imaging technology MICS. Deep phenotyping of infiltrating T cells was performed for the characterization of T cell activation or exhaustion states using the ultra-high-content MACSima™ Imaging Platform. Ex vivo analysis was performed in freshly excised, FFPE-treated tumor samples and compared for CD335 ESMA CAR treatment (n = 1) vs. the untransduced (UnTd) control (n = 1). (**a**) Composite images of DAPI (light blue), T cell markers CD4 (magenta), and CD8 (dark blue) with activation or exhaustion markers CD134 (yellow), PD-1 (orange), and Lag-3 (green), respectively. (**b**) Intra-tumoral distribution of total cells, CD4+ or CD8+ T cells, and ErbB2+ tumor cells within the CD335 ESMA CAR treatment group. The tumor was divided into inner, border, and outer regions as described in Appendix A. (**c**) Combined violin and swarm plots show the z-score-normalized intensity expression of T cell markers in three distinct tumor regions on CD4+ or CD8+ T cells, respectively.

**Figure 5 ijms-25-00615-f005:**
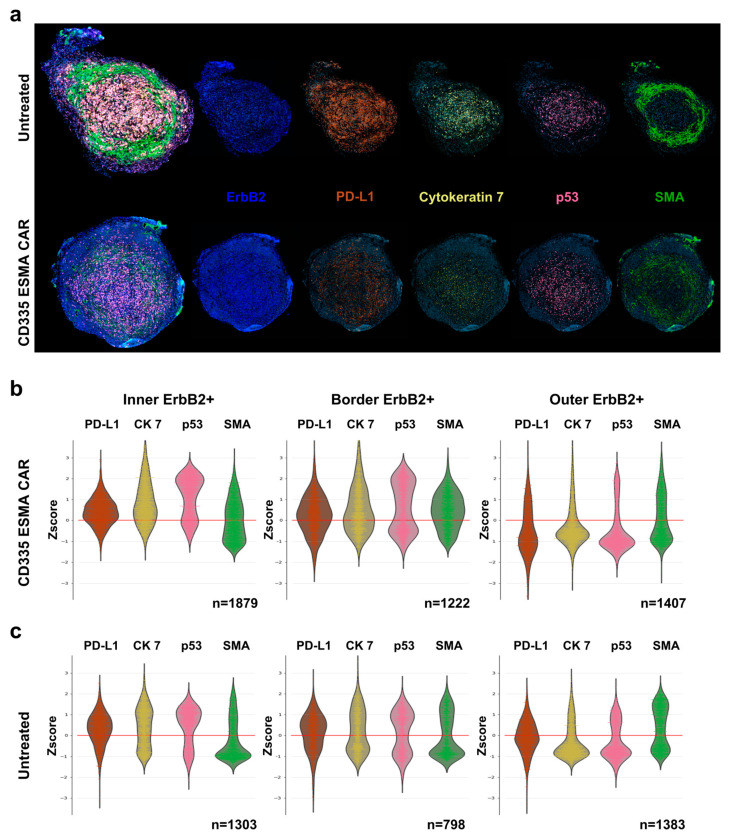
Intra-tumoral spatial analysis and phenotyping of tumor cells using the cyclic immunofluorescent imaging technology MICS. Deep phenotyping of tumor cells was performed to characterize tumor cell distribution and structural integrity as well as expression of relevant markers for T cell interaction using the ultra-high-content MACSima™ Imaging Platform. Ex vivo analysis was performed in freshly excised, FFPE-treated tumor samples and compared for CD335 ESMA CAR treatment (n = 1) vs. the untransduced (UnTd) control (n = 1). (**a**) Composite images of DAPI (light blue), tumor markers ErbB2 (dark blue) and PD-L1 (orange) with structural proteins cytokeratin 7 (CK 7, yellow) and smooth muscle actin (SMA, green), and apoptosis marker p53 (magenta). (**b**,**c**) Combined violin and swarm plots show the z-score-normalized intensity expression of tumor cell markers in three distinct tumor regions. The tumor was divided into an outer, border, and inner region as detailed in Appendix A.

## Data Availability

Data are contained within the article and Appendix A.

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
