# Peer review of "Endogenous Signaling Molecule Activating (ESMA) CARs: A Novel CAR Design Showing a Favorable Risk to Potency Ratio for the Treatment of Triple Negative Breast Cancer"

_ijms, 2024, doi:10.3390/ijms25010615_

Round 1

Reviewer 1 Report

Comments and Suggestions for Authors

This manuscript is interesting to read, however, the authors might consider the following points:- 

It will be ideal for the authors to assess the target specificity of ESMA CAR T cells. Use a panel of TNBC cell lines with varying antigen expression levels and non-TNBC cell lines as controls. Evaluate the cytotoxic activity and cytokine secretion of ESMA CAR T cells against these different cell lines to demonstrate the specificity of the ESMA CAR design. Additionally, assess potential off-target effects by examining the cytotoxic activity of ESMA CAR T cells against normal cells or non-specific antigens.

The Authors can also conduct a comparative analysis between the ESMA CAR design and traditional CAR designs (e.g., 2nd generation CARs) using in vitro and in vivo assays. Evaluate the cytotoxic activity, proliferation, cytokine secretion, exhaustion marker expression, and tumor infiltration of ESMA CAR T cells compared to traditional CAR T cells. This will help demonstrate the unique advantages and improved performance of the ESMA CAR design.

Finally, I would recommend performing experiments combining ESMA CAR T cells with standard chemotherapeutic agents or targeted therapies commonly used in TNBC treatment. Assess the cytotoxic activity, proliferation, and cytokine secretion of ESMA CAR T cells in combination with these therapies to determine if there are additive or synergistic effects. This will provide insights into the potential clinical applications of ESMA CAR T cells in combination therapies.

Discuss potential concerns or challenges associated with the implementation of ESMA CAR T cell therapy in clinical settings. Address issues such as safety, long-term persistence, scalability, and manufacturing feasibility. Provide insights into how these challenges could be overcome or mitigated.

Reviewer 2 Report

Comments and Suggestions for Authors

Manuscript IJMS-2775208, entitled “Endogenous signaling molecule activating (ESMA) CARs: A novel CAR design showing a favorable risk to potency ratio for the treatment of triple negative breast cancer” and prepared by Mira Ebbinghaus , Katharina Wittich , Benjamin Bancher , Valeriia Lebedeva , Anijutta Appelshoffer , Julia Femel , Martin Sebastian Helm , Jutta Kollet , Olaf Hardt, Rita Pfeifer, has been reviewed.

In this manuscript, an alternative chimeric antigen receptor (CAR) is reported that incorporates transmembrane domains with the ability to recruit endogenous signaling molecules, eliminating the need for stimulatory signals within the CAR structure. CAR T cell therapy is of general interest since it continues to gain attention as a valuable treatment option against different cancers, strategies for potency improvement and attempts for side effects reductions associated with this therapy. In principle, this is an interesting manuscript that could be of interest to a general audience. The manuscript is well written and the methodology they acquired is nicely described. Furthermore, the authors have presented their data clearly. Overall, I feel the manuscript could be accepted in the present form.

Minor comment:

1.      In line 57, tweak can be corrected.    

2.      In lines 150-151, only is repeated and could be replaced by another word.

Comments on the Quality of English Language

Minor editing of English is required.
